# A robust Th-azole framework for highly efficient purification of $C_2H_4$ from a $C_2H_4/C_2H_2/C_2H_6$ mixture

Zhenzhen Xu[1,5], Xiaohong Xiong[1,5], Jianbo Xiong[1], Rajamani Krishna[2], Libo Li [3], Yaling Fan[1], Feng Luo [1✉] & Banglin Chen [4✉]

Separation of $C_2H_4$ from $C_2H_4/C_2H_2/C_2H_6$ mixture with high working capacity is still a challenging task. Herein, we deliberately design a Th-metal-organic framework (MOF) for highly efficient separation of $C_2H_4$ from a binary $C_2H_6/C_2H_4$ and ternary $C_2H_4/C_2H_2/C_2H_6$ mixture. The synthesized MOF Azole-Th-1 shows a UiO-66-type structure with fcu topology built on a $Th_6$ secondary building unit and a tetrazole-based linker. Such noticeable structure, is connected by a N,O-donor ligand with high chemical stability. At 100 kPa and 298 K Azole-Th-1 performs excellent separation of $C_2H_4$ (purity > 99.9%) from not only a binary $C_2H_6/C_2H_4$ (1:9, v/v) mixture but also a ternary mixture of $C_2H_6/C_2H_2/C_2H_4$ (9:1:90, v/v/v), and the corresponding working capacity can reach up to 1.13 and 1.34 mmol $g^{-1}$, respectively. The separation mechanism, as unveiled by the density functional theory calculation, is due to a stronger van der Waals interaction between ethane and the MOF skeleton.

[1] State key Laboratory of Nuclear Resources and Environment, School of Chemistry, Biology and Material Science, East China University of Technology, 330013 Nanchang, P. R. China. [2] Van't Hoff Institute for Molecular Sciences, University of Amsterdam, Science Park 904, 1098 XH Amsterdam, The Netherlands. [3] College of Chemistry and Chemical Engineering, Shanxi Key Laboratory of Gas Energy Efficient and Clean Utilization, Taiyuan University of Technology, 030024 Taiyuan, Shanxi, China. [4] Department of Chemistry, University of Texas at San Antonio, San Antonio, TX, USA. [5]These authors contributed equally: Zhenzhen Xu, Xiaohong Xiong. ✉email: ecitluofeng@163.com; banglin.chen@utsa.edu

Ethylene is one of the most widely used feedstock molecules for the production of polymers and high-value organic chemicals[1,2]. It is usually produced by the thermal cracking of hydrocarbons. The removal of ethane and acetylene by-products that inevitably arise during these processes is one of the most challenging chemical separations due to the similarity of the physicochemical properties of ethane (kinetic diameter 4.4 Å, boiling point 184.55 K), ethylene (kinetic diameter 4.2 Å, boiling point 169.42 K), and acetylene kinetic diameter 3.3 Å, boiling point 188.40 K)[3–7].

At present, cryogenic distillation is the main technology used to separate ethane and ethylene with the requirement of high pressure (5–28 bar) and low temperature (180–258 K)[8,9], which indicates that this process is expensive and comes with a high energy penalty. And partial hydrogenation of acetylene into ethylene over catalyst[10] or solvent extraction of cracked olefins[11] are also involved with the purification of ethylene from acetylene. Adsorptive separation by porous materials is an alternative technology, especially, some metal-organic frameworks (MOFs)[12–21] with high volume, designable pore characteristics, and countless structural possibilities, can be employed into the gas separation processes, the adsorption selectivity and capacity are higher than the results of conventional adsorbents[22–24] such as zeolites and carbon-based, especially the adsorption and separation for $C_2H_6/C_2H_4$[8,22,25–39].

For the MOFs with open metal site, the ethylene can easily bind to it, leading to highly selective uptake of ethylene over ethane, due to the electrostatic interaction between the π-electron in ethylene and the positive charge in open metal sites[36,40–44], such as, HKUST-1[44], particularly in the low pressure region, is preferential adsorption of ethylene, which is supported by some theoretical calculations[45,46]. In contract, for some special MOFs, when the coordination positions of metal reaches to saturate, they can enable favorable adsorption towards ethane over ethylene through their unique pore wall that affords stronger van der Waals (vdW) interaction between the H of $C_2H_6$ and the MOF skeleton[8,22,25–29,32,34,35,47]. For example, ZIF-7 presents the first example of a microporous solid displaying the selective adsorption of paraffins over olefins[48]. However, MOFs showing such uncommon adsorption phenomenon ($C_2H_6$ over $C_2H_4$) is still rare until now.

Recently, Lu and co-workers[49] report that they use TJT-100 to obtain the selective adsorption of ethane and acetylene over ethylene from a ternary mixture of $C_2H_2/C_2H_6/C_2H_4$ (0.5:0.5:99, v/v/v) and achieve a $C_2H_4$ purity greater than 99.9% (working capacity of 0.69 mmol g$^{-1}$) by a single-breakthrough operation. Zaworoko et al.[38] use a synergistic sorbent separation method for the one-step production of polymer-grade $C_2H_4$ from ternary ($C_2H_2/C_2H_6/C_2H_4$, working capacity of 0.32 mmol g$^{-1}$) or quaternary ($CO_2/C_2H_2/C_2H_6/C_2H_4$) gas mixtures with a series of physisorbents. In this regard, constructing MOFs with high working capacity is still highly desirable from the viewpoint of practical application.

The high-valence metal ions are often used to construct stable MOFs, such as Cr(III) for MIL-101[50] and Zr(VI) for UiO-66[51]. However, due to the easy-to-hydrolysis nature of both Cr(III) and Zr(IV), it is still difficult to synthesize Cr(III) and Zr(IV) MOFs with high crystallization. Alternatively, another high-valence metal ion of Th(IV) shows less hydrolytic nature, suggesting an optimal metal ion to generate stable MOFs[52–56]. Generally speaking, according to the hard and soft acid and bases (HSAB) principle, the Zr-based or Th-based MOFs are usually constructed by O-donor carboxylic ligands. For example, the typical UiO-66-type structure is connected by various linear O-donor carboxylic ligands, as shown in the left of Supplementary Fig. 1[51,56,57]. As we know, the N,O-donor ligands such as azole

series have been attested to be an excellent organic linkers to construct a great number of MOFs (more than 900 tetrazole-based MOFs and more than 5000 triazole-based MOF from CCDC data)[58–60]. However, there is no Zr-based or Th-based MOFs built on N,O-donor ligands. As shown in the right of Supplementary Fig. 1, the construction of azole-based Zr- or Th-based MOFs is possible, because of the comparable coordination direction for carboxylate and azole molecules, while the introduction of azole unit to bind with Zr or Th ions is also possibly a powerful tool to modulate the electronic structure of metal center and the environment of pore wall, consequently leading to unique physical properties.

In this work, we obtain a successful case via solvothermal reaction of $Th(NO_3)_4$ and 4-(1H-tetrazol-5-yl) benzoic acid (TBA). This compound shows a UiO-66-type structure, except for the replacement of secondary building unit such as $Zr_6$ by $Th_6$ and linkers such as O-donor ligand by N,O-donor ligand. This unique tetrazole-based structure allows it to perform high $C_2H_6$ uptake at room temperature and selective adsorption of $C_2H_6$ over $C_2H_4$, finally resulting in the promising application for $C_2H_4$ separation from the binary $C_2H_6/C_2H_4$ and ternary $C_2H_4/C_2H_2/C_2H_6$ mixture. Furthermore, both the grand canonical Monte Carlo (GCMC) simulations and density functional theory (DFT) calculations are carried out to disclose the separation mechanism.

## Results

**Synthesis, structure, and characterization of Azole-Th-1.** The reaction between ligand TBA and $Th(NO_3)_4$ in N,N′-dimethyl-formamide (DMF) yields colorless octahedral crystals of Azole-Th-1 (Supplementary Fig. 2). The synthesis in detail is listed in Supplementary Information. The purity of the bulk samples was confirmed by powder X-ray diffraction (PXRD, Supplementary Fig. 3).

The PXRD discloses that the octahedral crystals of Azole-Th-1 crystallize in the cubic space group $Fm3m$, similar to UiO-66 (Fig. 1c). The length of $a$, $b$, $c$ is 23.984(4) Å, longer than UiO-66-Zr (20.743 (5) Å)[51] and UiO-66-Th (21.961(13) Å)[56], mainly due to the longer linker of TBA (ca. 8.4 Å) in Azole-Th-1 than terephthalic acid (ca. 6.8 Å) in UiO-66. Six Th(IV) metal ions are combined together to give the $Th_6O_4(OH)_4(H_2O)_6$ core (Fig. 1b), similar to the $Zr_6O_4(OH)_4(H_2O)_6$ in UiO-66. The Th(IV) ion holds the nine-coordination surrounding with a monocapped square antiprismatic geometry. The Th-O bond length from $O^{2-}$, $OH^-$, and $H_2O$ is varied from 2.33 to 2.55 Å, slight shorter than the Th–N bond length of 2.74 Å. As observed in the literature[61,62] for the TBA ligand usually showing highly disordered structure, similar trend is observed in Azole-Th-1 (Supplementary Fig. 4). The TBA ligand contacts with Th(IV) ions via both carboxylate and azole bridge, while two additional nitrogen atoms for each TBA ligand are free-standing without coordination. The 3D framework is formed by both the inorganic $Th_6O_4(OH)_4(H_2O)_6$ core and TBA linkers, where each inorganic core connects to twelve identical $Th_6O_4(OH)_4(H_2O)_6$ cores via twelve TBA linkers, finally constructing the UiO-66-like structure. Similarly, two different types of cages, viz. a super tetrahedron cage (Fig. 1d) and a super octahedron (Fig. 1e) with the largest cavity diameter of 1.1 nm and 1.2 nm, respectively, is observed in Azole-Th-1. This is larger the corresponding values of 0.88 nm in UiO-66[51], due to the longer linkers of TBA in Azole-Th-1 than terephthalic acid in UiO-66. The solvent-accessible volume estimated by Platon program[63] is 50.1% of the unit cell, suggesting high porosity of this MOF.

The loss of trapped solvent molecules from Azole-Th-1, according to the thermogravimetric (TG) analysis, is before 75 °C (Supplementary Fig. 5). While the temperature increased to

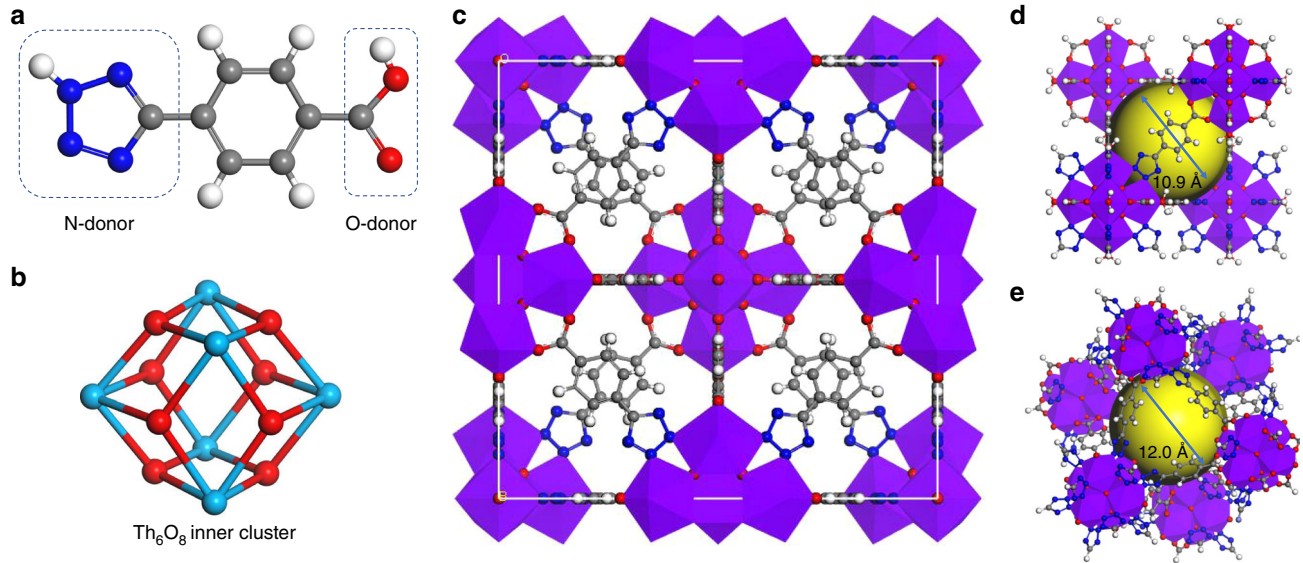

**Fig. 1 Main structures. a** Ligand TBA, including carboxylic acid donor (O-donor) and tetrazole donor (N-donor), **b** inner core Th$_6$-cluster drawn alone for clarity (Th$_6$O$_8$), **c** unit cell structure in crystal Azole-Th-1, **d**, **e** two types of cages, including super tetrahedron cage **d** and super octahedron cage **e**. Where, Th-light blue ball **b** and polyhedron style **c–e** with purple color, O-red ball, C-gray ball, N-blue ball, and H-white ball.

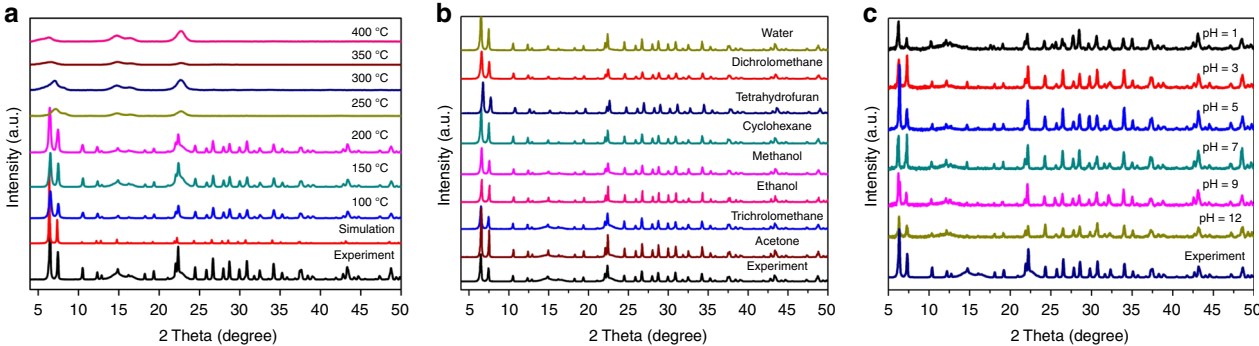

**Fig. 2 The PXRD patterns of Azole-Th-1 samples. a** Thermal stability from 100 to 400 °C, including simulated and experimental results, **b** soaking in water and seven different organic solvents 30 days, and **c** soaking in different pH solvents 30 days. Source data are provided as a Source Data file.

about 250 °C, the crystal structure begins to collapse, which is in agreement with the results of temperature dependent PXRD tests (Fig. 2a). As shown in Fig. 2a, the PXRD of Azole-Th-1 samples from room temperature to 200 °C are matching with the experimental and simulated results. While the temperature reaches to 250 °C, the peaks of PXRD disappear, which indicates that the crystal structure is destroyed by such high temperature. On the other hand, the chemical stability under water and different solvents environment, including seven organic solvents and a broad pH range from 1 to 12, were also traced by PXRD tests (Fig. 2b, c), where respective optical images of crystal were represented in Fig. 3. Note that the crystals of Azole-Th-1 render excellent stability in water and above solvents even after 30 days.

**Adsorption isotherm, selectivity, and breakthrough**. The moderate thermal stability and high chemical stability of Azole-Th-1 prompts us to study the gas adsorption performances. To characterize the permanent porosity of the obtained material, the N$_2$ adsorption isotherm at 77 K was measured. As shown in Fig. 4a, a fully reversible type I isotherm with a Brunauer Emmett Teller (BET) surface area of 983 m$^2$ g$^{-1}$ and a uniform pore size around 9.2 Å was exhibited. This pore size is comparable

with that calculated LCD (the largest cavity diameter, 10.0 Å) by Zeo++ program[64] (Supplementary Table 3).

This high porosity and desirable aperture encouraged us to further investigate C$_2$H$_6$/C$_2$H$_4$ separations in detail. Adsorption isotherms of single component C$_2$H$_6$ and C$_2$H$_4$ were collected at 298 K and 273 K, respectively, as presented in Fig. 4b and Supplementary Fig. 6. The adsorption isotherm of C$_2$H$_6$ is typically type I with a steep slope, which is a typical feature of strong adsorbates in microporous materials (Fig. 4b)[8,22,25,27–30,32–35]. And the adsorption amounts of C$_2$H$_6$ at both 273 and 298 K (121.7 and 100.2 cm$^3$ g$^{-1}$) were also higher than the corresponding C$_2$H$_4$ (111.3 and 80.7 cm$^3$ g$^{-1}$). Therefore, the Azole-Th-1 has a distinct preference for adsorbing ethane over ethylene. It is well-known that the magnitude of the adsorption enthalpies of porous materials reveals that the affinity of the pore surface toward adsorbents, determining the adsorptive selectivity[39,65]. This can be directly reflected on the adsorption heat enthalpy ($Q_{st}$) of C$_2$H$_6$ and C$_2$H$_4$, giving 28.6 kJ mol$^{-1}$ for C$_2$H$_6$ at the zero coverage, significantly higher than the values of 26.1 kJ mol$^{-1}$ for C$_2$H$_4$ (Fig. 4d), strongly suggesting a higher affinity between host and guest for C$_2$H$_6$ than C$_2$H$_4$, where the detail virial-type analysis were provided in Supplementary Fig. 7.

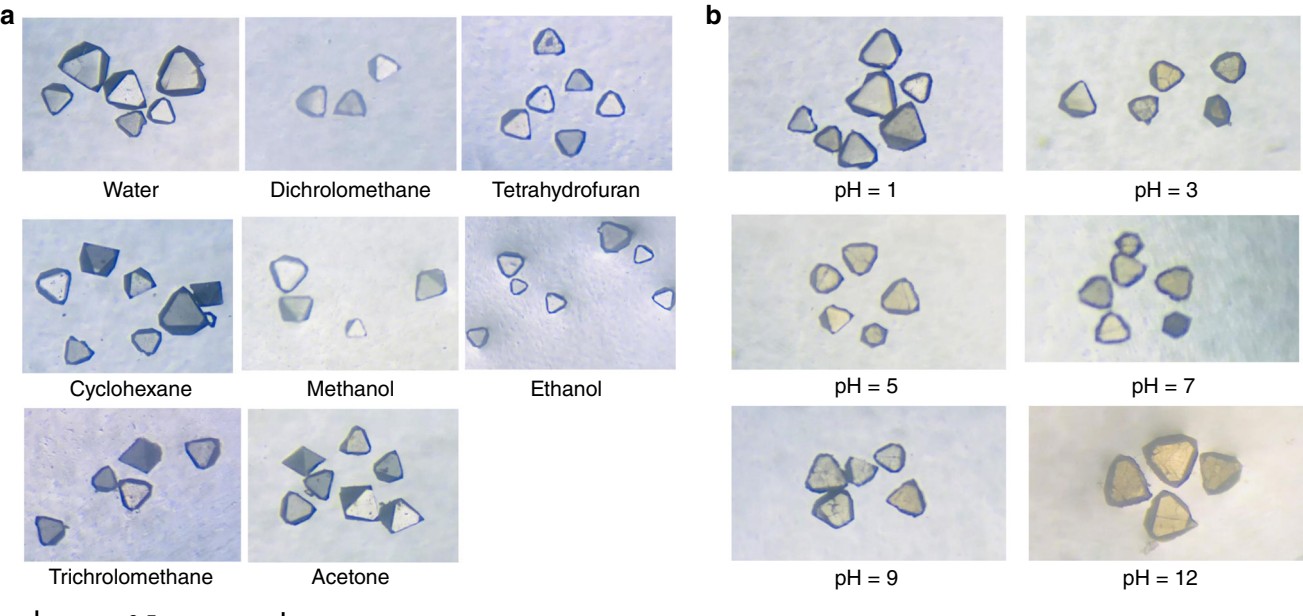

**Fig. 3 The optical microscope images of Azole-Th-1 samples. a** after soaking in water and seven different organic solvents 30 days, **b** after soaking in different pH solvents 30 days.

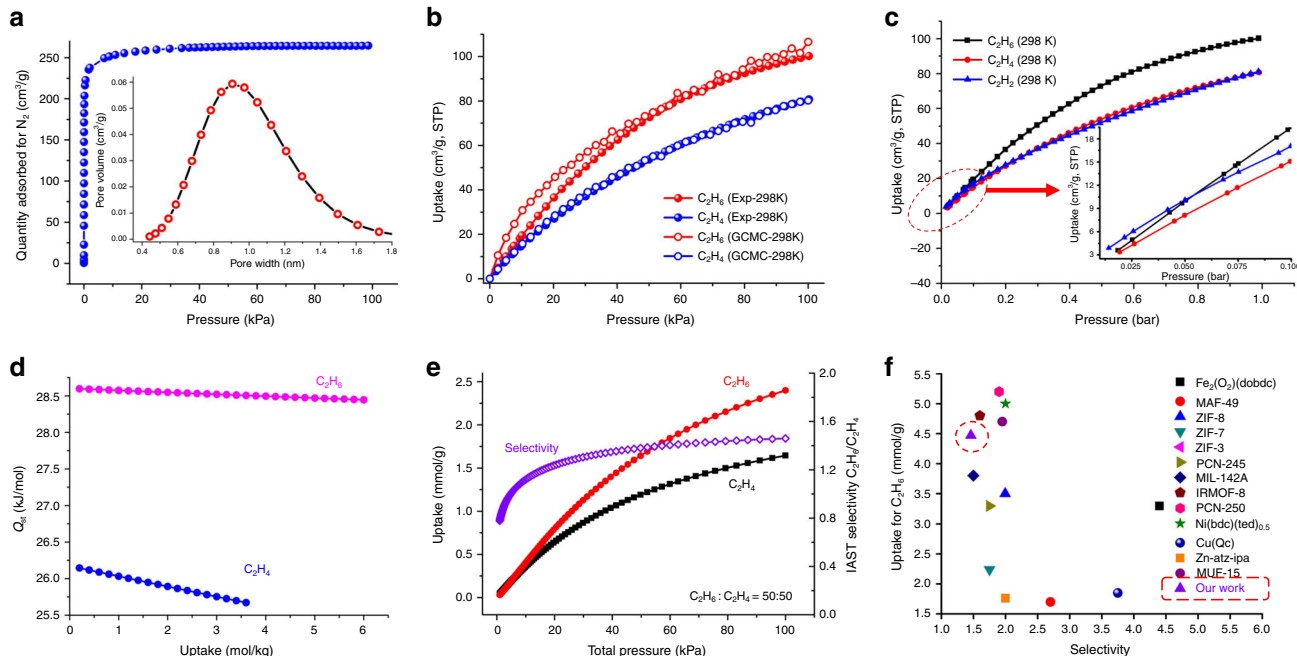

**Fig. 4 The adsorption and separation data of Azole-Th-1. a** The $N_2$ adsorption at 77 K with the insert of the distribution of pore size. **b** The adsorption isotherms of $C_2H_6$ and $C_2H_4$ at 298 K, including experiments and simulations. **c** Experimental adsorption isotherms of Azole-Th-1 for $C_2H_6$, $C_2H_4$, and $C_2H_2$ at 298 K from 0.01 to 1 bar with the insert of enlargement from 0.01 to 0.1 bar. **d** The adsorption heat enthalpy of $C_2H_6$ and $C_2H_4$, calculated from the single-component $C_2H_6$ and $C_2H_4$ adsorption data at 298 and 273 K. **e** Predicted mixture adsorption isotherms and selectivity of Azole-Th-1 by IAST method for a 50/50 $C_2H_6/C_2H_4$ mixture at 298 K. **f** A comparison in selectivity and $C_2H_6$ adsorption capacity at 298 K and 1 bar between the reported top-performing porous adsorbents for $C_2H_6/C_2H_4$ separation and our MOF. The purple triangle is our MOF. Source data are provided as a Source Data file.

At room temperature and 100 kPa, Azole-Th-1 affords ultrahigh adsorption capacity of $C_2H_6$ up to 100.2 cm³ g⁻¹. This value exceeds most reported top-performing porous adsorbents for such use as shown in Table S2, including in $Fe_2(O_2)(dobdc)$[35], MAF-49[22], ZIF-8[34], ZIF-7[32], PCN-245[26], MIL-142A[28], Cu $(Qc)_2$[37], Zn-atz-ipa[38], etc., which are summarized in Fig. 4f. Correspondingly, adsorption capacity of $C_2H_4$ at the same

conditions is 81.1 cm³ g⁻¹, obviously, less than $C_2H_6$ about 20 cm³ g⁻¹. Hence, selective adsorption of $C_2H_6$ over $C_2H_4$ is suggested. To estimate the adsorption selectivity, we employed the ideal adsorption solution theory (IAST)[66] to analyze the experimental isotherm data, using composition of 50:50/10:90/1:15 $C_2H_6/C_2H_4$, as shown in Fig. 4e and Supplementary Fig. 8. The selectivity of $C_2H_6$ over $C_2H_4$ was up to 1.46 at room

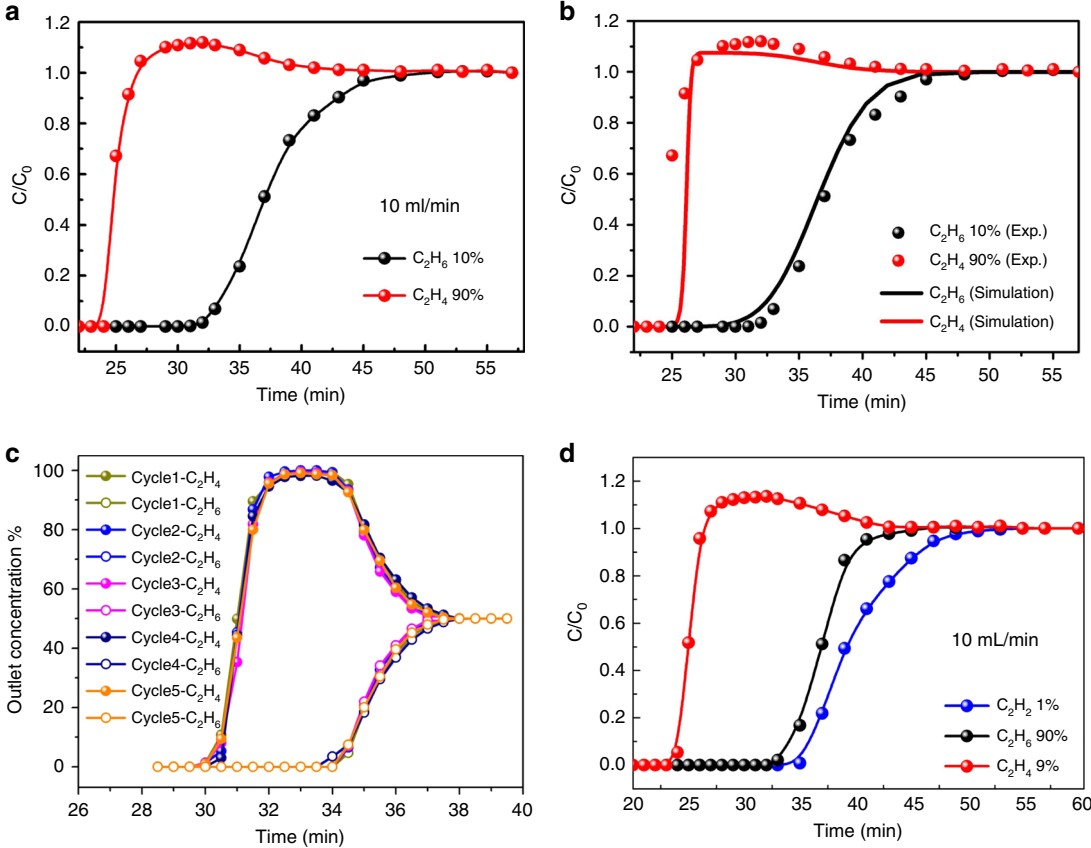

**Fig. 5 Experimental breakthrough curves at 298 K and 1 bar on Azole-Th-1. a**, **b** $C_2H_6/C_2H_4$ (10/90, v/v) binary mixture. **c** $C_2H_6/C_2H_4$ (50/50, v/v) binary mixture for five cycles. **d** $C_2H_6/C_2H_4/C_2H_2$(90/9/1, v/v/v) ternary mixture separation. Source data are provided as a Source Data file.

temperature and 100 kPa, which is slightly higher than that of other ratio mixture, both 1.44 for 10:90 and 1:15 $C_2H_6/C_2H_4$. To the best of our knowledge, Azole-Th-1 should be the first Th-MOF showing such abnormal adsorption behavior ($C_2H_6$ over $C_2H_4$).

Then, in order to check the actual separation ability of the gases mixture, the transient breakthrough simulations for $C_2H_6/C_2H_4$ (50/50, v/v) mixtures on Azole-Th-1 material was carried out at 298 K, as shown in Supplementary Fig. 9. This result demonstrates that the potential of producing pure product gas $C_2H_4$ during the time interval $\Delta\tau$, also suggests excellent $C_2H_6/C_2H_4$ separation performance. And then, to further confirm the real practical separation ability, the actual dynamic adsorption breakthrough experiments for $C_2H_6/C_2H_4$ (10/90, 50/50, 1/15, v/v) binary mixtures on Azole-Th-1 material were also carried out (Fig. 5a, b, and Supplementary Fig. 10). The $C_2H_4$ broke through the adsorption bed and yield a high purity gas (>99.9%) at first, whereas after a certain time $C_2H_6$ slowly eluted and reached to the equilibrium (Fig. 5a, b, and Supplementary Fig. 10). During this period of time, polymer-grade (>99.9%) $C_2H_4$ can be generated at the outlet. The breakthrough time of ethane was later than that of ethylene for these three ratio mixtures, meaning that the Azole-Th-1 preferred to adsorb ethane over ethylene. The long breakthrough time interval between $C_2H_4$ and $C_2H_6$ suggests that the Azole-Th-1 is quite effective for $C_2H_6/C_2H_4$ separation. The experimental breakthrough results was well in consistent with the simulated breakthrough (Fig. 5b, Supplementary Fig. 10b, d), strongly suggesting its superior application for $C_2H_4$ purification. Furthermore, cycling breakthrough experiments on Azole-Th-1 were carried out under the same conditions. The breakthrough time interval for $C_2H_6/C_2H_4$ mixtures in five cycles

(Fig. 5c) is comparable, showing that this material has a good regenerability. According to the polymer grade $C_2H_4$ produced during time interval at different $C_2H_4/C_2H_6$ ratio, 3 min (50/50), 5 min (90/10), and 3.5 min (15/1), the productivities of $C_2H_4$ (>99.9%) were 0.68, 1.13, and 0.79 mmol g$^{-1}$, respectively. Hence, the polymer grade $C_2H_4$ with the max working capacity of 1.13 mmol g$^{-1}$ with >99.9% purity was harvested from 90/10 gas mixture, which working capacity is nearly 1.3 times for $Fe_2(O_2)$ (dobdc) (0.79 mmol g$^{-1}$)[35] and 3.6 times for MAF-49 (0.28 mmol g$^{-1}$)[22], the two best materials for $C_2H_6/C_2H_4$ separation. Some more detailed comparison with other MOF materials is shown in Supplementary Table 2. After the breakthrough experiments, the PXRD pattern of our sample was also consistent with the PXRD before the breakthrough (Supplementary Fig. 11), which further indicated that this material has a good regenerability and high stability.

**Mechanism of gas adsorption by theoretical calculations.** Theoretically, the determination of gas adsorption sites in the MOFs is of great significance for the design of some gas storage and separation materials based on MOFs[31,67]. Herein, the ultrahigh $C_2H_6$ storage capacity prompts us to explore the adsorption sites within this Azole-Th-1. Theoretical simulation is a powerful tool enabling us to unveil the adsorption mechanisms and provide the adsorption sites. Therefore, the $C_2H_6/C_2H_4$ binding affinity in Azole-Th-1 was firstly investigated by single-component sorption isotherms at 298 K and pressures up to 100 kPa. The GCMC simulations were performed for understanding the interactions and adsorption behaviors of $C_2H_6$ and $C_2H_4$ in Azole-Th-1 at the molecular level[25,26,33,34,39]. As shown in isotherm of $C_2H_6/C_2H_4$

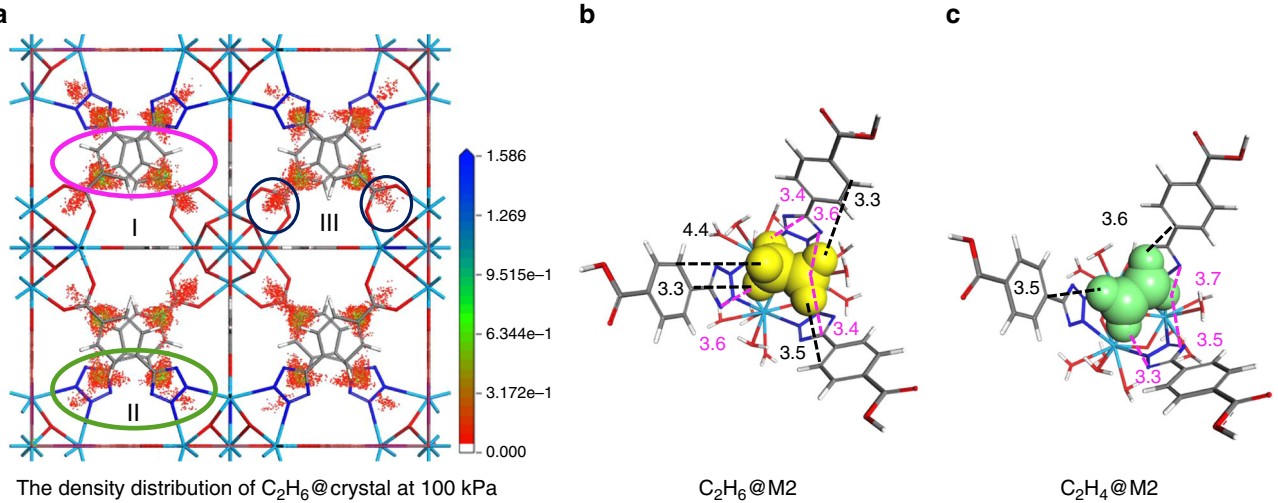

**Fig. 6 GCMC adsorption simulation and adsorbed structures. a** The density distribution of $C_2H_6$ on Azole-Th-1 at 100 kPa and 298 K. **b, c** The structures of adsorptions for $C_2H_6$ and $C_2H_4$ at M2 model. Where, Th-light blue, O-red, C-gray, N-blue, H-white, $C_2H_6$-yellow molecule, and $C_2H_4$-green molecule, and the unit of distance is Å.

adsorption (Fig. 4b), the maximum $C_2H_6$ and $C_2H_4$ uptake for Azole-Th-1 are 106.6 and 80.2 $cm^3\,g^{-1}$ at 100 kPa and 298 K, respectively, both adsorbed tendency and the adsorption quantity are consistent with the experimental results (100.2 and 80.7 $cm^3\,g^{-1}$), where the simulated details are listed in Supplementary Information.

The further investigations on the interaction between $C_2H_6$/$C_2H_4$ and MOF material can help us to understand the mechanism of gas adsorption, which could analysis the discrepancies the interactions between $C_2H_6$ and $C_2H_4$ with our material, respectively. According to the density distribution of $C_2H_6$ on Azole-Th-1 at 298 K and 100 kPa (Fig. 6), there are three main adsorbed areas in our material, including benzene region (I-region), the tetrazol heterocycle region (II-region), and carboxylate region (III-region), respectively. Then, the adsorptions under different pressure were analyzed (Supplementary Fig. 12). Due to the higher polarizability of $C_2H_6$ ($44.7 \times 10^{-25}\,cm^3$) compared with $C_2H_4$ ($42.5 \times 10^{-25}\,cm^3$)[7], at the beginning of the adsorption, the $C_2H_6$ molecules are preferentially filled in region I at 2.6 kPa (Supplementary Fig. 12a), but the $C_2H_4$ molecules are almost not adsorbed until 10.3 kPa (Supplementary Fig. 12e). While the pressure is 5.1 kPa, the region II begins to be filled by $C_2H_6$ molecules (Supplementary Fig. 12b), the total capacity of adsorption reaches to 18.5 $cm^3\,g^{-1}$, which corresponds to the adsorption capacity of $C_2H_4$ at 12.8 kPa (18.9 $cm^3\,g^{-1}$), at that moment the region II is empty until the pressure increases to 17.9 kPa (Supplementary Fig. 12f). With the increase of pressure (to 48.7 kPa), region I and region II are almost filled saturated by $C_2H_6$ molecules (Supplementary Fig. 12c), and region III begins to be filled by $C_2H_6$ already, the total capacity of adsorption reaches to 73.2 $cm^3\,g^{-1}$. The vdW interactions between $C_2H_6$ with aromatic rings (regions I and II) and carboxylate (region III) are more remarkable than the interactions between $C_2H_4$ with them. While the pressure reaches to 100 kPa, these three regions are saturated generally by $C_2H_6$ (Supplementary Fig. 12d), whereas, the $C_2H_4$ molecules only continue to be filled into regions I and II (Supplementary Fig. 12g), the $C_2H_6$ uptake (106.2 $cm^3\,g^{-1}$) is significantly greater than the maximum $C_2H_4$ uptake (80.2 $cm^3\,g^{-1}$). Hence, according to the GCMC simulations, the Azole-Th-1 prefers to adsorbing the $C_2H_6$ from 0.001 to 100 kPa at 298 K, and the $C_2H_6$ adsorption capacity is far greater than $C_2H_4$, which could achieve the separation of $C_2H_6$ and $C_2H_4$.

And then, according to the obtained adsorption regions, some DFT theoretical calculations about the mechanism of selective $C_2H_6$/$C_2H_4$ in Azole-Th-1 were investigated by the Dmol$^3$ program package[68] in the MS, the detail calculations were also presented in Supplementary Information. The determination of the adsorption point can quantify the interaction between the gas $C_2H_6$/$C_2H_4$ and Azole-Th-1 and analyze the mechanism of the gas adsorption. Because the calculations using the whole unit cell is too large, we used the fragmented cluster models cleaved from the unit cell for modeling the structures and energies to investigate the interaction points of $C_2H_6$/$C_2H_4$ adsorption. Due to the highly disordered structure of TBA, four fragment models were constructed. Accordingly, the fragment M1 to M4 were intercepted (Supplementary Fig. 13). Based on the distribution of density of $C_2H_6$/$C_2H_4$ through GCMC simulations, the geometries of fragmented models bound to $C_2H_6$/$C_2H_4$ were also obtained, as shown in Supplementary Fig. 14. By the calculations of binding energy for adsorbed geometries (Supplementary Table 5), obviously, the interactions between M1 to M4 with $C_2H_6$ ($-43.09\,kJ\,mol^{-1}$) were stronger than the interactions between that of fragmented models with $C_2H_4$ ($-33.52\,kJ\,mol^{-1}$), which results were in agreement with the heat of adsorption $Q_{st}$. These can be attributed to the vdW interaction between the C–H in $C_2H_6$/$C_2H_4$ and conjugated π-systems in Azole-Th-1, especially, the conjugated region II (tetrazol).

In our opinion, there are three factors for the stabilities of adsorbed structures, the number of H atoms, the distances of vdW interaction, and the polarizability, where the summary about previous two factors were listed in Supplementary Tab. 6. As shown in Supplementary Fig. 14, $C_2H_6$ and $C_2H_4$ molecules are bound to several aromatic rings of ligands at three directions within a pore through vdW interactions. A single $C_2H_6$ or $C_2H_4$ molecule can form six or four pairs of C–H—π interactions with conjugated regions at least, where the vdW interaction between C–H and benzene ring of ligand (region I) are in the majority. The greater the number of H atoms, the stronger the C–H—π interaction between gas with different models, and finally the greater the adsorption capacity of $C_2H_6$ compared with $C_2H_4$.

Because of the use of previously unreported ligand, the conjugated region tetrazole can enhance the binding with gas. As shown in Supplementary Fig. 14, $C_2H_6$@M1 (all O-donors of ligands coordinated on the Th(IV) inner cluster), there are two

pairs of vdW interactions between C–H with carboxylic acid (region III). However, as to $C_2H_6$@M2 (Fig. 6, all N-donor of ligands coordinated on the Th(IV) inner cluster), because the gas molecule was near to the region II, which contributed to the increasing of the interaction probability between gas and tetrazol. The four pairs of C–H—$\pi$ interactions with region II could make the adsorption structure more stable with $-46.90\,\mathrm{kJ\,mol^{-1}}$ binding energy than the structure without the interaction between $C_2H_6$ with tetrazol ($C_2H_6$@M1, $-25.87\,\mathrm{kJ\,mol^{-1}}$).

And then, the average distances between $C_2H_6$ with benzene region (I), tetrazol region (II), and carboxylic acid region (III) are 3.67, 3.65, and 3.08 Å, respectively. And the average distances between $C_2H_4$ with these three regions are 3.41 and 3.25 Å (no interaction with region III), which results agreed with the GCMC results (Supplementary Fig. 12g, $C_2H_4$ only filled into regions I and II). According to these distances of interaction, it is hard to analysis the stabilization of the adsorption structures. Besides the distance and number of C–H—$\pi$ interactions, $C_2H_6$ as a more polarizable molecule can interact more strongly by induced dipole interactions with the framework compared to the less polarizable $C_2H_4$ molecule.

**Separation of ternary mixture of $C_2H_6/C_2H_2/C_2H_4$.** In traditional $C_2H_4$ production process, trace amounts of $C_2H_2$ (about 1%) also exist in the ethylene feed. So, this material was further investigated for the simultaneous capture of $C_2H_2$ and $C_2H_6$ from a ternary mixture of $C_2H_6/C_2H_2/C_2H_4$. As shown in Fig. 5d, highly efficient separation of $C_2H_4$ from a 90:1:9 (v/v/v) gas mixture of $C_2H_6/C_2H_2/C_2H_4$ was achieved by passing the mixture over a packed column of activated Azole-Th-1 material. It can be observed that $C_2H_4$ achieves a breakthrough first, with no evidence of $C_2H_2$ or $C_2H_6$ flow before its breakthrough, which indicates that this material can produce an high purity $C_2H_4$ (>99.9%) after only a single-breakthrough operation. The working capacity is up to $1.34\,\mathrm{mmol\,g^{-1}}$, far exceeding the top-performing materials reported by Lu et al. ($0.69\,\mathrm{mmol\,g^{-1}}$)[49] and Zaworoko et al. ($0.32\,\mathrm{mmol\,g^{-1}}$)[38], strongly suggesting its promising applications in this target. Note that, according to the adsorption isotherm of $C_2H_2$ (Fig. 4c), the maximum of capacities of $C_2H_2$ (81.1 cm³ g⁻¹) and $C_2H_4$ (80.7 cm³ g⁻¹) are almost equal to each other. In addition, the adsorption heat enthalpy of $C_2H_2$ is $25.4\,\mathrm{kJ\,mol^{-1}}$ at the zero coverage, which is lower slightly than $Q_{st}$ of $C_2H_4$ ($26.1\,\mathrm{kJ\,mol^{-1}}$). And why the high purity of $C_2H_4$ (>99.9%) can be acquired. Therefore, in order to explore this problem, the $C_2H_6$, $C_2H_4$, and $C_2H_2$ at low pressure region (<0.1 bar) was checked (inserted into Fig. 4c). Before pressure of 0.05 bar, the adsorption of $C_2H_2$ is obviously bigger than both $C_2H_6$ and $C_2H_4$, giving a hierarchy of $C_2H_2 > C_2H_6 > C_2H_4$. This is well consistent with the separation conditions of 90:1:9 (v/v/v) gas mixture of $C_2H_6/C_2H_2/C_2H_4$.

## Discussion

In a word, we reported a Th-azole framework (Azole-Th-1) by introducing an previously unreported ligand TBA showing a preferential adsorption of ethane over ethylene. Notably, Azole-Th-1 samples exhibited good stability for soaking in water, various organic solvents, and different pH (1–12) solvents about 30 days, respectively. Moreover, preferential adsorption of ethane over ethylene was confirmed by measuring the adsorption isotherms and breakthrough curves. Azole-Th-1 had relative high ethane and ethylene adsorption capacities, 4.5 and 3.6 mmol g⁻¹ at 298 K and 100 kPa, respectively. The adsorption selectivity of binary mixture $C_2H_6/C_2H_4$ (1:1, v/v) was ~1.46 at pressure below 100 kPa and 298 K. Five cycles of ethane adsorption–desorption cycle experiments revealed that Azole-Th-1 had a good regenerability. The

polymer grade $C_2H_4$ with the max working capacities of 1.13 mmol g⁻¹ with 99.9% purity was harvested from 10/90 gas mixture. Furthermore, Azole-Th-1 also can purify the $C_2H_4$ (purity >99.9%) from ternary mixture $C_2H_6/C_2H_2/C_2H_4$ (90:1:9, v/v/v) with working capacities of 1.34 mmol g⁻¹. Some DFT calculations suggested that the greater vdW interaction between ethane and Azole-Th-1 than ethylene and material, $-43.09$ and $-33.52\,\mathrm{kJ\,mol^{-1}}$, respectively, which were in agreement with the isosteric heat of ethane ($28.6\,\mathrm{kJ\,mol^{-1}}$) and ethylene ($26.1\,\mathrm{kJ\,mol^{-1}}$). In brief, these excellent properties, the perfect pH stability and high $C_2H_4$ purity (>99.9%) from a ternary 90:1:9 mixture of $C_2H_6/C_2H_2/C_2H_4$, etc., make Azole-Th-1 a promising candidate for efficient separation of ethane/ethylene. It will be much more challenging and difficult to separate more complex gas mixtures. Those small gas molecules such as $N_2$ and $CH_4$ will not affect the ternary $C_2H_6/C_2H_2/C_2H_4$ separation very much because of their very weak interactions with the framework, leading to very low uptakes of $N_2$ and $CH_4$ at the room temperature. However, some other gas molecules, particularly $CO_2$, is expected to significantly affect the $C_2H_6/C_2H_2/C_2H_4$ separation, because the uptakes of $CO_2$ are comparable to these C2 hydrocarbons. Until now, no suitable porous materials have been reported yet for the efficient $C_2H_6/C_2H_2/C_2H_4/CO_2$ separation. Before any porous materials can be realized for this very challenging separation, step-by-step separations by different adsorbents will be necessary to get high purity C2 hydrocarbons.

## Methods

**The synthesis**. A mixture of 4-(1H-Tetrazol-5-yl) benzoic acid (TBA, 0.019 g, 0.10 mmol), Th(NO₃)₄ (0.048 g, 0.10 mmol), N,N′-dimethylformamide (DMF, 3.0 mL), and ionic liquid of tetramethylguanidine chloride (0.015 mg, 0.1 mmol) were placed in a 20 mL screw-capped glass capped jar, then five drops of concentrated hydrochloric acid were added to the mixture. The mixture was sealed and heated at 110 °C for 3 days. The reaction system was cooled to 30 °C with about 6 °C per min cooling rate. After filtration and washed with excess of N,N′-dimethylacetamide (DMA), colorless block crystals were collected as a pure phase (see PXRD in Fig. 2).

**Gas-adsorption and breakthrough experiments**. The original sample about 100 mg was activated at 60 °C under high vacuum for 12 h in gas adsorption apparatus before the gas adsorption measurement. The BET of the MOFs were investigated by nitrogen adsorption and desorption at 77 K using a Belsorp-max. The single-component isotherms of $C_2H_6$, $C_2H_4$, and $C_2H_2$ were collected at 298 and 273 K on a Belsorp-max. The breakthrough separation apparatus consisted of two fixed-bed stainless steel column. One column was loaded with MOF powder (1.9810 g), while the other reactor was used as a blank control group to stabilize the gas flow. The horizontal reactors were placed in a temperature-controlled environment, maintained at 298 K. The flow rates of all gases mixtures were regulated by mass flow controllers, and the effluent gas stream from the column is monitored by a gas chromatography (TCD-Thermal Conductivity Detector, detection limit 0.1%). Prior to each breakthrough experiment, we regenerated the sample by flushing the adsorption bed with helium gas (100 mL per min) for 30 min at 298 K.

## Data availability

The X-ray crystallographic coordinated for structure reported in this study has been deposited at the Cambridge Crystallographic Data Centre (CCDC), under deposition number 1969398. This data can be obtained free of charge from the CCDC via https://www.ccdc.cam.ac.uk/structures/. The source data underlying Figs. 2, 4, and 5 and Supplementary Figs. 5, 6, 8, 9, 10a, 10c, and 11 are provided as a Source Data file. And other data, if not included in the article or Supplementary Information or Source Data, are available from the authors on request.

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

## Acknowledgements

We thanks to the National Natural Science Foundations of China (21966002, 21871047, 21661001, and 21922810), the Natural Science Foundation of Jiangxi Province of China (20181ACB20003), and the Training Program for Academic and Technical Leaders of Major Disciplines in Jiangxi Province (20194BCJ22010).

## Author contributions

B.C. and F.L. conceived and designed the research, and gave valuable comments on the analysis; L.L. carried out the transient breakthrough experiments; R.K. calculated the selectivity and simulated the breakthrough; Z.X. carried out all theoretical calculations and interpretations, created all figures and wrote the draft; X.X., J.X., and Y.F. carried out experiments.

## Competing interests

The authors declare no competing interests.
