## [Peer Review File · Nature Communications]

Reviewers' comments:

Reviewer #1 (Remarks to the Author):

This manuscript reports on the separation of ethane from ethylene using a metal-organic framework. This is a challenging task using conventional media, so absorptive separations on new kinds of porous media are a hot topic. This manuscript presents a new material that accomplishes this separation fairly well. The framework itself is inherently of some interest since it is built up from Th(IV) ions, which is a rather uncommon approach. Importantly, the material displays 'inverted' selectivity i.e. it adsorbs ethane in preference to ethylene. The manuscript is well presented and illustrated and is easy to follow and extract the key messages.

Therefore, following the serious consideration of the following points, this manuscript is probably suitable for publication in Nature Comms.

- MUF-15 has been left out off Figure 6f but its data should be presented here.
- Table S2 should be referred to around L192.
- Why are Fe₂(O)₂(dobdc) and MAF-49 chosen for the comparisons of productivity (L222)? Since these materials have low uptake their productivity may not be expected to be high. Comparisons with other materials would be more appropriate.
- Please provide the code used for the breakthrough simulations. This is very important so that other researchers can verify these calculation. Literature citations are given but these do no contain the code itself.
- The actual Clausius-Clapeyron plot used to calculate the Q_{st} values should be presented.
- What temperature is used for Table S2?
- In Figure S2, "simulated by" should be "simulated from".
- In the caption to Fig S6, the definition of tau should be stated. When does it start and when does it stop? This probably related to the composition of the outlet gas stream, and should be stated clearly.
- There is a grammatical error on L261/262.
- On L107, UIO-66 should be UiO-66.
- The reported BET surface area is overly precise (2 decimal places!). 983 would be better.

Reviewer #2 (Remarks to the Author):

This is a very well written piece of interesting work that certainly deserves to be published. However, I believe that Nature Communications is not the best journal for it. First, because it is not clear that individual MOFs such as Azole-Th-1 are interesting for the general audience. In addition, the authors recently published another quite similar article, but applied to another MOF, with title "Robust Microporous Metal-Organic Frameworks for the highly efficient and simultaneous removal of propylene and propylene from propylene" *Angewandte Chemie* 131 (30). As I said, this is a good job, but it should

be published in a more specialized journal.

Reviewer #3 (Remarks to the Author):

This work from Banglin Chen and coworkers details the C₂H₄ purification performance of a new Th(IV) MOF that is based upon an azolate carboxylate mixed N,O-donor ligand. In my opinion, the results are of high topical relevance to adsorptive purification and the data and conclusions presented are of a high standard. I therefore support publication after the authors address the following comments.

1. What is the molecular formula of Azole-Th-1 ?
2. Fig. 1 could be moved to the Supplementary information as the first figure of SI because the coordination modes reported, although previously unreported, are expected based upon the literature on Zr-MOFs and Th-MOFs.
3. The TGA data for Azole-Th-1 needs to be remeasured using a slower ramp rate. Currently, the authors write "The loss of trapped solvent molecules from Azole-Th-1, according to the thermogravimetric (TG) analysis, is before 100 °C (Supplementary Figure 4)". However, Figure 4 in SI shows that there is no stable plateau region for Azole-Th-1, even after 100 C. This conclusion is also inconsistent with the temperature dependent PXRD data (Fig. 3a).
4. The optical microscope images need to include scale bars (Figures 4 and 5); also the authors can possibly combine Figures 4 and 5.
5. Whereas the SI states that "The reagents and solvents were commercially available and were used as received without further purification", I would recommend that the authors include more details on which commercial resource / vendors / purity grade were used to procure the tetrazole carboxylic acid.
6. The synthesis section mentions "The reaction system was cooled to 30°C." After a solvothermal reaction, cooling rate is often critical to afford good quality single crystals as the authors have obtained. This information should be included.
7. What is the role of ionic liquid tetramethylguanidine chloride in the synthesis? What happens if this reagent is excluded?
8. The abbreviation "UiO-66" has been misquoted as "UIO-66" needs to be corrected all over the manuscript and SI.
9. I would recommend the authors to avoid using the term "unprecedented" for a new ligand (included twice in the manuscript). I suggest the use of more suitable alternatives such as "previously unreported", "new" etc.
10. This story is exciting on two levels with respect to properties: first, the excellent pH stability; second, achieving C₂H₄ purification (purity > 99.9%) from a ternary 9:1:90 mixture of C₂H₆/C₂H₂/C₂H₄. I would appreciate it if the authors could discuss future outlooks in the Discussion section. In particular, could the new Th-MOF be useful for more complex gas mixtures including those containing CO₂ or would it have to be used synergistically with other sorbents.

Reviewer #1 Comments:

This manuscript reports on the separation of ethane from ethylene using a metal-organic framework. This is a challenging task using conventional media, so absorptive separations on new kinds of porous media are a hot topic. This manuscript presents a new material that accomplishes this separation fairly well. The framework itself is inherently of some interest since it is built up from Th(IV) ions, which is a rather uncommon approach. Importantly, the material displays 'inverted' selectivity i.e. it adsorbs ethane in preference to ethylene. The manuscript is well presented and illustrated and is easy to follow and extract the key messages.

Therefore, following the serious consideration of the following points, this manuscript is probably suitable for publication in Nature Comms.

Q1. MUF-15 has been left out off Figure 6f but its data should be presented here.

Response: Thank this reviewer for the very constructive comment.

This is our negligence, the data for MUF-15 has been added into the Figure 6f.

The original Figure 6f

The modified Figure 4f

Q2. Table S2 should be referred to around L192.

Response:

At L192, Table S2 has been referred with yellow highlight.

The original:

'This value exceeds most reported top-performing porous adsorbents for such use including in Fe₂(O₂)(dobdc)³⁷, MAF-49²¹, ZIF-8³⁵, ZIF-7^{29, 30, 33}, PCN-245²⁵, MIL-142A²⁷, Cu(Qc)₂³⁹, Zn-atz-ipa⁴⁰, etc. (Fig. 6f).'

Modified to :

'This value exceeds most reported top-performing porous adsorbents for such use as shown in Table

S2, including in $\text{Fe}_2(\text{O}_2)(\text{dobdc})$ ³⁷, MAF-49²¹, ZIF-8³⁵, ZIF-7^{29, 30, 33}, PCN-245²⁵, MIL-142A²⁷, $\text{Cu}(\text{Qc})_2$ ³⁹, Zn-atz-ipa⁴⁰, etc., which are summarized in Fig. 4f.'

Q3. Why are $\text{Fe}_2(\text{O})_2(\text{dobdc})$ and MAF-49 chosen for the comparisons of productivity (L222)? Since these materials have low uptake their productivity may not be expected to be high. Comparisons with other materials would be more appropriate.

Response:

Thank you very much for pointing out this.

Productivity is dependent on several parameters, particularly selectivity, gas uptake and pore volume. At present, the reported top-performing MOFs for productivities of C_2H_4 are $\text{Fe}_2(\text{O})_2(\text{dobdc})$ with 0.79 mmol/g (>99.99%), MAF-49 with 0.28 mmol/g (>99.99%), ZIF-7 with 0.01 mmol/g (>99%), ZIF-8 with 0.03 mmol/g (>99%), IRMOF-8 with 0.12 mmol/g (>99%), and PCN-250 with 0.15 mmol/g (>99%). Indeed, $\text{Fe}_2(\text{O})_2(\text{dobdc})$ and MAF-49 are two best materials for the $\text{C}_2\text{H}_6/\text{C}_2\text{H}_4$ separation and thus chosen to compare with our material Azole-Th-1.

The original:

'Hence, the polymer grade C_2H_4 with the max working capacity of 1.13 mmol/g with > 99.9% purity was harvested from 90/10 gas mixture, which working capacity is nearly 1.3 times for $\text{Fe}_2(\text{O}_2)(\text{dobdc})$ (0.79 mmol/g)³⁷ and 3.6 times for MAF-49 (0.28 mmol/g)²¹.'

Modified to :

'Hence, the polymer grade C_2H_4 with the max working capacity of 1.13 mmol/g with > 99.9% purity was harvested from 90/10 gas mixture, which working capacity is nearly 1.3 times for $\text{Fe}_2(\text{O}_2)(\text{dobdc})$ (0.79 mmol/g)³⁷ and 3.6 times for MAF-49 (0.28 mmol/g)²¹, the two best materials for $\text{C}_2\text{H}_6/\text{C}_2\text{H}_4$ separation. Some more detailed comparison with other MOF materials is shown in Supplementary Tab. 2.'

Q4. Please provide the code used for the breakthrough simulations. This is very important so that other researchers can verify these calculation. Literature citations are given but these do no contain the code itself.

Response:

Thanks for reviewer's comment.

The numerical details and breakthrough code implementation are provided online: Krishna, R.; Baur, R. Diffusion, Adsorption and Reaction in Zeolites: Modelling and Numerical Issues.

<http://krishna.amsterchem.com/zeolite/>. University of Amsterdam, Amsterdam, 1 January 2015.

And in Supplementary Information of article, the related reference has been cited in Section 9 of Supplementary Information.

The original:

'Simulations The performance of industrial fixed bed adsorbers is dictated by a combination of adsorption selectivity and uptake capacity. Transient breakthrough simulations were carried out for 50/50, 90/10, and 15/1 binary C₂H₄(1)/C₂H₆(2) mixtures and 9/1/90 ternary C₂H₆/C₂H₂/C₂H₄ mixture in **Azole-Th-1** operating at a total pressure of 100 kPa and 298 K, using the methodology described in earlier publications.^{12, 13, 14, 15} For the breakthrough simulations, the following parameter values were used: length of packed bed, $L = 0.3$ m; voidage of packed bed, $\varepsilon = 0.4$; superficial gas velocity at inlet, $u = 0.04$ m/s.'

Modified to :

'Simulations The performance of industrial fixed bed adsorbers is dictated by a combination of adsorption selectivity and uptake capacity. Transient breakthrough simulations were carried out for 50/50, 90/10, and 15/1 binary C₂H₄(1)/C₂H₆(2) mixtures and 9/1/90 ternary C₂H₆/C₂H₂/C₂H₄ mixture in **Azole-Th-1** operating at a total pressure of 100 kPa and 298 K, using the methodology described in earlier publications.^{12, 13, 14, 15} **The numerical details of the code implementation are provided online by Krishna and Baur.**¹⁶ For the breakthrough simulations, the following parameter values were used: length of packed bed, $L = 0.3$ m; voidage of packed bed, $\varepsilon = 0.4$; superficial gas velocity at inlet, $u = 0.04$ m/s.'

Q5. The actual Clausius-Clapeyron plot used to calculate the Q_{st} values should be presented.

Response:

Thanks for reviewer's comment.

For the 1-site Langmuir-Freundlich isotherm, the Q_{st} can be calculated analytically by analytic differentiation of the Clausius-Clapeyron equation. We have provided the actual Clausius-Clapeyron plot in Figure **S7** of Supplementary Information, as follow,

Equation	$y = \ln(x) + 1/K * (a_0 + a_1 * x + a_2 * x^2 + a_3 * x^3 + a_4 * x^4 + a_5 * x^5) + (b_0 + b_1 * x + b_2 * x^2)$		
Reduced Chi-Sqr		4.99989E-4	
Adj. R-Square		0.99964	
	Value	Standard Error	
B	a0	-3440.2764	7.6824
	a1*	3.36302	0.93853
	a2*	-0.04332	0.0227
	a3*	8.47576E-4	3.34532E-4
	a4*	-7.69124E-6	2.14462E-6
	a5*	2.40871E-8	4.9957E-9
	b0*	12.65212	0.02118
	b1*	-0.01189	0.00295
	b2*	7.08512E-5	2.53916E-5
	K	273	0

(a)

Model	Qstfinal (User)		
Equation	$y = \ln(x) + 1/K * (a_0 + a_1 * x + a_2 * x^2 + a_3 * x^3 + a_4 * x^4 + a_5 * x^5) + (b_0 + b_1 * x + b_2 * x^2)$		
Reduced Chi-Sqr		8.28979E-6	
Adj. R-Square		0.99999	
	Value	Standard Error	
B	a0	-3148.3776	10.8867
	a1*	17.29921	0.26691
	a2*	-0.13963	0.00465
	a3*	6.5908E-4	7.61471E-5
	a4*	-4.7508E-6	5.78482E-7
	a5*	1.57899E-8	1.59597E-9
	b0*	11.62162	0.00355
	b1*	-0.04569	8.57449E-4
	b2*	3.11847E-4	7.85013E-6
	K	273	0

(b)

Model	Qstfinal (User)		
Equation	$y = \ln(x) + 1/K * (a_0 + a_1 * x + a_2 * x^2 + a_3 * x^3 + a_4 * x^4 + a_5 * x^5) + (b_0 + b_1 * x + b_2 * x^2)$		
Reduced Chi-Sqr		0.0018	
Adj. R-Square		0.99878	
	Value	Standard Error	
B	a0	-3058.544	9.2246
	a1*	-5.88164	2.26404
	a2*	0.0955	0.05987
	a3*	3.19415E-4	0.00116
	a4*	-1.39815E-6	9.05101E-6
	a5*	5.92879E-9	2.56536E-8
	b0*	11.02903	0.0407
	b1*	0.05061	0.00841
	b2*	-5.82205E-4	1.05417E-4
	K	273	0

(c)

Supplementary Figure 7. The fitting curves for Q_{st} of C_2H_6 (a), C_2H_4 (b), and C_2H_2 (c) by Clausius-Clapeyron equation between 273 K and 298K.

Q6. What temperature is used for Table S2?

Response:

Thanks for reviewer's comment.

All temperatures in Table S2 are 298K, we have added into the title of Table S2.

The original Table:

Supplementary Table 2. A summary of reported porous adsorbents for C₂H₆/C₂H₄ separation.

	C ₂ H ₆ /C ₂ H ₄ Uptakes (mmol/g) at 100 kPa	C ₂ H ₆ /C ₂ H ₄ Selectivity (50/50)	Q _{st} (kJ/mol)
Fe ₂ (O ₂)(dobdc) ¹⁰	3.3/2.5	4.4	66.8/38
MAF-49 ¹⁷	1.70/1.65	2.7	60/48
IRMOF-8 ¹⁸	4.8/3.4	1.6	52.5/50.5
ZIF-8 ¹⁹	3.5/1.8	1.99	22.2/16.3
ZIF-7 ^{20, 21, 22}	2.24/2.2	1.75	27.3/24.7
ZIF-3 ¹⁹	6.0/5.5	2.22	28.5/23.8
PCN-250 ²³	5.2/4.1	1.9	23.6/21.1
Ni(bdc)(ted) _{0.5} ²⁴	5/3.2	2.0	21.5/18.3
MUF-15 ²⁵	4.7/4.2	1.95	29.2/28.2
PCN-245 ²⁰	3.3/2.4	1.75	23/20.5
MIL-142A ²⁷	3.8/2.9	1.5	27.3/26.2
Cu(Qc) ₂ ²⁸	1.85/0.78	3.75	29/25.4
Zn-atz-ipa ²⁹	1.76/1.75	2.0	45.8/40
Our MOF	4.5/3.6	1.46	27.1/24.8

Modified to :

Supplementary Table 2. A summary of reported porous adsorbents for C₂H₆/C₂H₄ separation at 1 bar and 298 K.

	C ₂ H ₆ /C ₂ H ₄ Uptakes (mmol/g)	C ₂ H ₆ /C ₂ H ₄ Selectivity (50/50)	Q _{st} (kJ/mol)
Fe ₂ (O ₂)(dobdc) ¹⁷	3.3/2.5	4.4	66.8/38
MAF-49 ¹⁸	1.70/1.65	2.7	60/48
IRMOF-8 ^{17, 19}	4.8/3.4	1.6	52.5/50.5
ZIF-8 ²⁰	3.5/1.8	1.99	22.2/16.3
ZIF-7 ^{21, 22, 23}	2.24/2.2	1.75	27.3/24.7
ZIF-3 ²⁰	6.0/5.5	2.22	28.5/23.8
PCN-250 ²⁴	5.2/4.1	1.9	23.6/21.1
Ni(bdc)(ted) _{0.5} ²⁵	5/3.2	2.0	21.5/18.3
MUF-15 ²⁶	4.7/4.2	1.95	29.2/28.2
PCN-245 ²⁷	3.3/2.4	1.75	23/20.5
MIL-142A ²⁸	3.8/2.9	1.5	27.3/26.2
Cu(Qc) ₂ ²⁹	1.85/0.78	3.75	29/25.4
Zn-atz-ipa ³⁰	1.76/1.75	2.0	45.8/40
Our MOF	4.5/3.6	1.46	27.1/24.8

Q7. In Figure S2, "simulated by" should be "simulated from".

Response:

Thanks for reviewer's comment.

We have modified according to the reviewer's proposal.

The original:

‘ **Supplementary Figure 2.** The PXRD patterns involved in this work, including the as-synthesized samples and the simulation by single crystal.’

Modified to :

‘**Supplementary Figure 3.** The PXRD patterns involved in this work, including the as-synthesized samples and the simulation **from** single crystal.’

Q8. In the caption to Fig S6, the definition of tau should be stated. When does it start and when does it stop? This probably related to the composition of the outlet gas stream, and should be stated clearly.

Response:

Thanks for reviewer’s comment.

The definition of the dimensionless time is already provided in section 9 of Supplementary information. The time $t=0$, corresponds to feed injection. The calculations are continued until the bed reaches equilibrium.

Q9. There is a grammatical error on L261/262.

Response:

Thanks for reviewer’s comment.

The original:

‘Due to the higher polarizability of C_2H_6 ($44.7 \times 10^{-25} \text{ cm}^3$) compared with C_2H_4 ($42.5 \times 10^{-25} \text{ cm}^3$)⁷, at the beginning of the adsorption, the C_2H_6 molecules are preferentially filled in region I at 2.6 kPa (Supplementary Fig. 9a), but the C_2H_4 molecules are almost no adsorption until 10.3 kPa (Supplementary Fig. 9e).’

Modified to :

‘Due to the higher polarizability of C_2H_6 ($44.7 \times 10^{-25} \text{ cm}^3$) compared with C_2H_4 ($42.5 \times 10^{-25} \text{ cm}^3$)⁷, at the beginning of the adsorption, the C_2H_6 molecules are preferentially filled in region I at 2.6 kPa (Supplementary Fig. 12a), but the C_2H_4 molecules are almost not adsorbed until 10.3 kPa (Supplementary Fig. 12e).’

Q10. On L107, UIO-66 should be UiO-66.

Response:

Thanks for reviewer’s comment.

All 13 ‘UIO-66’ are modified to UiO-66 with **yellow highlight** in main thesis.

Q11. The reported BET surface area is overly precise (2 decimal places!). 983 would be better.

Response:

Thanks for reviewer's comment.

We have done a modification about this problems.

The original:

'As shown in Fig. 5a, a fully reversible type I isotherm with a Brunauer Emmett Teller (BET) surface area of 982.97 m²/g and a uniform pore size around 9.2 Å was exhibited.'

Modified to :

'As shown in Fig. 4a, a fully reversible type I isotherm with a Brunauer Emmett Teller (BET) surface area of 983 m²/g and a uniform pore size around 9.2 Å was exhibited.'

Reviewer #2 Comments:

This is a very well written piece of interesting work that certainly deserves to be published. However, I believe that Nature Communications is not the best journal for it. First, because it is not clear that individual MOFs such as Azole-Th-1 are interesting for the general audience. In addition, the authors recently published another quite similar article, but applied to another MOF, with title "Robust Microporous Metal-Organic Frameworks for the highly efficient and simultaneous removal of propylene and propylene from propylene" *Angewandte Chemie* 131 (30). As I said, this is a good job, but it should be published in a more specialized journal.

Response:

Thank you very much for your very positive comments. It appears that this reviewer misunderstood and was thus confused with the difference between this work and the one this reviewer mentioned [30]. For the clear comparison, we made the table shown below. It is very clear that these two works are completely different. More importantly, ref 30 is about the separation of propyne/propadiene/propylene; while this work is about the separation of ethylene/ ethane/ acetylene which has not been fully explored yet. To the best of our knowledge, this work is the best one for this ternary gas separation among the developed porous materials. I do humbly disagree with this reviewer and think that this work deserves the publication in *Nature Communications*.

Similarity		◇ MOFs; ◇ Sorption and separation; ◇ Ternary mixed gas	
Differences		Angew Chem Int Ed Engl	Our Works
	Name	NKMOF-1-M	Azole-Th-1
	Ligand	Pyrazine-2,3-dithiol	4-(1H-Tetrazol-5-yl) benzoic acid
	Metal	Cu(II) and Ni(II)	Th(IV)
	Coordinated Manner	Single Coordination	Mixed Coordination
	Purified Gas	The separation for C3 (propylene from ternary mixture of propyne/propadiene/propylene).	The separation for C2 (ethylene from ternary mixture of ethylene/ ethane/ acetylene).

Reviewer #3 Comments:

This work from Banglin Chen and coworkers details the C₂H₄ purification performance of a new Th(IV) MOF that is based upon an azolate carboxylate mixed N,O-donor ligand. In my opinion, the results are of high topical relevance to adsorptive purification and the data and conclusions presented are of a high standard. I therefore support publication after the authors address the following comments.

Q1. What is the molecular formula of Azole-Th-1 ?

Response:

Thank this reviewer for the very positive and constructive comments.

The molecular formula of Azole-Th-1 $\text{Th}_6\text{O}_4(\text{OH})_4(\text{H}_2\text{O})_6(\text{TBA})_6$ has been provided in 'Abstract', and the crystal data as well in Supplementary Tab. 1 for the framework only without the consideration of the guest solvents.

Q2. Fig. 1 could be moved to the Supplementary information as the first figure of SI because the coordination modes reported, although previously unreported, are expected based upon the literature on Zr-MOFs and Th-MOFs.

Response:

Thanks for reviewer's comment.

We have moved Fig.1 to Supplementary information as the first figure, named 'Supplementary Figure 1'.

Q3. The TGA data for Azole-Th-1 needs to be remeasured using a slower ramp rate. Currently, the authors write "The loss of trapped solvent molecules from Azole-Th-1, according to the thermogravimetric (TG) analysis, is before 100 °C (Supplementary Figure 4)". However, Figure 4 in SI shows that there is no stable plateau region for Azole-Th-1, even after 100 C. This conclusion is also inconsistent with the temperature dependent PXRD data (Fig. 3a).

Response:

Thanks for reviewer's comment.

According to the reviewer's suggestion, the TG analysis using 2°C/min ramp rate has been explored. From the new results of experiments, the loss of trapped solvent DMF from Azole-Th-1

sample is before 75°C, which is earlier than the original results using 10°C/min, but the plateau region is still descending slightly due to the incomplete loss of DMF molecules. However, the plateau region of sample soaking in methanol three days is more stable, which is the result of completely exchange between DMF and methanol.

And then, in order to verify the sample is always loss of DMF, the TG analysis for sample soaking in methanol one day is also tested. The plateau of loss solvent is between the sample and the sample soaked in methanol three days, which indicates that the solvent exchange between DMF and methanol is not complete. Hence, reduce the ramp rate to 2°C/min, it is always occurred for the loss of solvent, which is not improved the results well.

In our article, the corresponding modifications as follow,

The original in manuscript:

The loss of trapped solvent molecules from **Azole-Th-1**, according to the thermogravimetric (TG) analysis, is before 100 °C (Supplementary Fig. 4). While the temperature increased to about 300°C, the crystal structure begins to collapse, which is in agreement with the results of temperature dependent PXRD tests (Fig. 3a).

Modified to:

The loss of trapped solvent molecules from **Azole-Th-1**, according to the thermogravimetric (TG) analysis, is before 75 °C (Supplementary Fig. 5). While the temperature increased to about 250 °C, the crystal structure begins to collapse, which is in agreement with the results of temperature dependent PXRD tests (Fig. 2a).

The original in Section 7 of Supplementary Information:

All TGA experiments were performed under a N₂ atmosphere from 25 °C to 800 °C at a rate of 10 °C /min.

Modified to:

All TGA experiments were performed under a N₂ atmosphere from room temperature to 800 °C at a rate of 2 °C /min.

Q4. The optical microscope images need to include scale bars (Figures 4 and 5); also the authors can possibly combine Figures 4 and 5.

Response:

Thanks for reviewer's comment.

According to reviewer's suggestion, we have combined these two into one picture, named Figure 3, and the scale bar is add below this picture.

The original:

Fig. 4 The optical microscope images of **Azole-Th-1** samples after soaking in water and seven different organic solvents 30 days.

Fig. 5 The optical microscope images of **Azole-Th-1** samples after soaking in different pH solvents 30 days.

Modified to:

Fig. 3 The optical microscope images of **Azole-Th-1** samples after soaking in water and seven different organic solvents (a) and different pH solvents (b) 30 days.

Q5. Whereas the SI states that “The reagents and solvents were commercially available and were used as received without further purification”, I would recommend that the authors include more details on which commercial resource / vendors / purity grade were used to procure the tetrazole carboxylic acid.

Response:

Thanks for reviewer’s comment.

We have added some detailed information about the ligand tetrazole benzoic acid, other reagents, and solvents in section 1 of Supplementary Information.

The original:

‘The reagents and solvents were commercially available and were used as received without further purification.’

Modified to:

‘The reagents and solvents were commercially available and were used as received without further purification, where the ligand 4-(1H-Tetrazol-5-yl) benzoic acid ($C_8H_6N_4O_2$, 95%) from EXTENSION Technology Co., Ltd was purchased and directly used, Thorium nitrate hydrate ($N_4O_{12}Th$, Aladdin), Tetramethylguanidine chloride ($C_5H_{14}N_3Cl$, 98%, Aladdin), hydrochloric acid

(HCl, Aladdin), and solvents (N, N'-dimethylformamide and N, N'-dimethylacetamide, HPLC grade of 99.9%) from Aladdin Chemistry Co. Ltd were also purchased and directly used.'

Q6. The synthesis section mentions “The reaction system was cooled to 30°C.” After a solvothermal reaction, cooling rate is often critical to afford good quality single crystals as the authors have obtained. This information should be included.

Response:

Thanks for reviewer's comment.

During the holding process at 110 °C, the crystal has completed the growth process. After three days, the power of the oven is turned off, and it will cool with about 6 °C/min cooling rate to room temperature. The corresponding modification is also added in our manuscript.

The original:

'The reaction system was cooled to 30°C.'

Modified to:

'The reaction system was cooled to 30°C with about 6 °C/min cooling rate'

Q7. What is the role of ionic liquid tetramethylguanidine chloride in the synthesis? What happens if this reagent is excluded?

Response:

Thanks for reviewer's comment.

If this ionic liquid tetramethylguanidine is excluded, the crystal is hard to obtain in our experiments. In our opinion, this ionic liquid may act as a template in the synthesis. The further research on the affection of ionic liquid is in progress.

Q8. The abbreviation “UiO-66” has been misquoted as “UIO-66” needs to be corrected all over the manuscript and SI.

Response:

Thanks for reviewer's comment.

This is really our negligence for this misquotation, 'UIO-66' has been quoted 13 times in our manuscript, so all 13 'UIO-66' are modified to UiO-66 with yellow highlight.

Q9. I would recommend the authors to avoid using the term “unprecedented” for a new ligand (included twice in the manuscript). I suggest the use of more suitable alternatives such as “previously unreported”, “new” etc.

Response:

Thanks for reviewer's comment.

The original:

This term has been used **three** times, as follow,

1. 'Because of the use of unprecedented ligand, the conjugated region tetrazole can enhance the binding with gas.'
2. 'In a word, we firstly reported a Th-azole framework (**Azole-Th-1**) by introducing an unprecedented ligand TBA showing a preferential adsorption of ethane over ethylene.'
- 3.

Fig. 1 The modes of Zr(IV) or Th(IV) based MOFs using the carboxylic acid ligand (typical ligand) and azole ligand (unprecedented ligand), respectively.

Modified to:

1. 'Because of the use of **previously unreported** ligand, the conjugated region tetrazole can enhance the binding with gas.'
2. 'In a word, we firstly reported a Th-azole framework (**Azole-Th-1**) by introducing an **previously unreported** ligand TBA showing a preferential adsorption of ethane over ethylene.'
- 3.

Supplementary Figure 1. The modes of Zr(IV) or Th(IV) based MOFs using the carboxylic acid ligand (typical ligand) and azole ligand (**previously unreported ligand**), respectively.

Q10. This story is exciting on two levels with respect to its properties: first, the excellent pH stability; second, achieving C₂H₄ purification (purity > 99.9%) from a ternary 9:1:90 mixture of C₂H₆/C₂H₂/C₂H₄. I would appreciate it if the authors could discuss future outlooks in the Discussion section. In particular, could the new Th-MOF be useful for more complex gas mixtures including those containing CO₂ or would it have to be

used synergistically with other sorbents.

Response:

Thanks for reviewer's comment.

According to reviewer's suggestion, we have added the discussion about future outlooks at last paragraph.

The original:

'In brief, these excellent properties make **Azole-Th-1** a promising candidate for efficient separation of ethane/ethylene.'

Modified to:

'In brief, these excellent properties, the perfect pH stability and high C₂H₄ purity (> 99.9%) from a ternary 90:1:9 mixture of C₂H₆/C₂H₂/C₂H₄, etc., make **Azole-Th-1** a promising candidate for efficient separation of ethane/ethylene. It will be much more challenging and difficult to separate more complex gas mixtures. Those small gas molecules such as N₂ and CH₄ will not affect the ternary C₂H₆/C₂H₂/C₂H₄ separation very much because of their very weak interactions with the framework, leading to very low uptakes of N₂ and CH₄ at the room temperature. However, some other gas molecules, particularly CO₂, is expected to significantly affect the C₂H₆/C₂H₂/C₂H₄ separation, because the uptakes of CO₂ are comparable to these C₂ hydrocarbons. Until now, no suitable porous materials have been reported yet for the efficient C₂H₆/C₂H₂/C₂H₄/CO₂ separation. Before any porous materials can be realized for this very challenging separation, step-by-step separations by different adsorbents will be necessary to get high purity C₂ hydrocarbons.'

In addition, we also have some other modifications on our article, such as:

- 1. The last sentence of Fig. 1 has been done a simple modifications.**
- 2. The temperature unit 'K' in Fig. 2a is changed to '°C', and the corresponding modifications in article also have been done with the yellow highlight.**
- 3. The unit of pressure in Figure 4a, 4b, and 4e, Fig. S6, and Fig. S12 are modified to 'kPa' instead of 'KPa'.**
- 4. A little discussion about the selectivity of 10:90/1:15 C₂H₆/C₂H₄ are added to our article, and the corresponding figures are presented in Fig. S8.**
- 5. In Supplementary Information, we have added a safety caution about radioactivity at the beginning of Section 1.**

6. Some content of C₂H₂ has newly added into the Table S4.
7. The format of all tables in Supplementary Information have an adjustment.

Noted: all modifications are highlighted with yellow in our manuscript and Supplementary Information.

REVIEWER COMMENTS

Reviewer #1 (Remarks to the Author):

The changes have improved the manuscript substantially and I can no recommend acceptance subject to one point. The Q_{st} values are derived from the analysis presented in Supplementary Fig 7. However, this looks a virial analysis rather than the stated Clausius-Clapeyron.

For a Clausius-Clapeyron analysis, the isosteric heat of adsorption (DH) can be calculated as:

$$DH / R = (d \ln p) / (d 1/T)$$

So the plot in S7 should show $(\ln p)$ as a function of $(1/T)$ at the same loading amount.

So this all needs clarification and the actual equations and parameters should be presented in the ESI (in text form in addition to the plots). All the fitting lines should be shown on the plots too (some missing at the moment).

Reviewer #3 (Remarks to the Author):

The authors have satisfactorily address my comments and I recommend acceptance of this manuscript without further revision.

Reviewer #1 Comments:

The changes have improved the manuscript substantially and I can no recommend acceptance subject to one point. The Q_{st} values are derived from the analysis presented in Supplementary Fig 7. However, this looks a virial analysis rather than the stated Clausius-Clapeyron.

For a Clausius-Clapeyron analysis, the isosteric heat of adsorption (DH) can be calculated as:

$$DH / R = (d \ln p) / (d 1/T)$$

So the plot in S7 should show $(\ln p)$ as a function of $(1/T)$ at the same loading amount.

So this all needs clarification and the actual equations and parameters should be presented in the ESI (in text form in addition to the plots). All the fitting lines should be shown on the plots too (some missing at the moment).

Response: Thank this reviewer for the very constructive comment.

This is really our negligence for this confusion. The actual equation we using is viral-type equation, as follow,

$$\ln(p) = \ln(N) + \frac{1}{T} \sum_{i=0}^m a_i N_i + \sum_{i=0}^n b_i N_i$$

According to our adsorption data from experiments at 273 and 298 K, the parameters of series of a_i and b_i can be obtained by fitting curves, which results are provided in Supplementary Fig. 7.

And then, based on the determined parameters of series a_i by the viral analysis, the Q_{st} can be

obtained by equation $Q_{st} = -R \sum_{i=0}^m a_i N^i$. In our Supplementary Information, the actual equation has

been clarified in Section 4, which is shown in the below. The parameters for a_i and b_i have been presented in Figure S7, the missed fitting line is also added.

The modification 1:

(a)

(b)

(c)

Supplementary Figure 7. The virial equation fitting curves for C₂H₆ (a), C₂H₄ (b), and C₂H₂ (c) adsorption isotherms of Azole-Th-1.

The modification 2:

where p is the pressure, T is the temperature, R is the gas constant (8.314 J/mol·K). By drawing the $\ln p$ vs $1/T$ plot of gas at various loadings, $Q_{st} = -slope \times R$. To extract the coverage-dependent isosteric heat of adsorption, the data were modeled with a virial-type expression^{3,4} composed of parameters a_i and b_i that are independent of temperature:

$$\ln(p) = \ln(N) + \frac{1}{T} \sum_{i=0}^m a_i N^i + \sum_{i=0}^n b_i N^i \quad (2)$$

$$Q_{st} = -R \sum_{i=0}^m a_i N^i \quad (3)$$

Where N is the amount adsorbed (or uptake), m and n determine the number of terms required to adequately describe the isotherm. The isosteric heat of adsorption is calculated according to Eq. 3. The coverage dependencies of Q_{st} calculated from fitting the 273 and 298 K data are presented graphically in Fig. 4d, and the virial equation fit for C₂H₆, C₂H₄, and C₂H₂ adsorption isotherms of Azole-Th-1 are shown in Supplementary Fig. 7.